# Near-term forecasting of Covid-19 cases and hospitalisations in Aotearoa New Zealand

**Michael J. Plank** [1] *, **Leighton Watson** [1], **Oliver J. Maclaren** [2]

**1** School of Mathematics and Statistics, University of Canterbury, Christchurch, New Zealand, **2** Department of Engineering Science, University of Auckland, Auckland, New Zealand

* michael.plank@canterbury.ac.nz

## Abstract

Near-term forecasting of infectious disease incidence and consequent demand for acute healthcare services can support capacity planning and public health responses. Despite well-developed scenario modelling to support the Covid-19 response, Aotearoa New Zealand lacks advanced infectious disease forecasting capacity. We develop a model using Aotearoa New Zealand's unique Covid-19 data streams to predict reported Covid-19 cases, hospital admissions and hospital occupancy. The method combines a semi-mechanistic model for disease transmission to predict cases with Gaussian process regression models to predict the fraction of reported cases that will require hospital treatment. We evaluate forecast performance against out-of-sample data over the period from 2 October 2022 to 23 July 2023. Our results show that forecast performance is reasonably good over a 1-3 week time horizon, although generally deteriorates as the time horizon is lengthened. The model has been operationalised to provide weekly national and regional forecasts in real-time. This study is an important step towards development of more sophisticated situational awareness and infectious disease forecasting tools in Aotearoa New Zealand.

## Author summary

The emergency phase of the Covid-19 pandemic has ended, but Covid-19 continues to put significant additional load on stretched healthcare systems. Forecasting the number of hospital cases caused an infectious disease like Covid-19 over the next few weeks can help with effective planning and response. The ability to forecast reliably requires timely, high-quality data and accurate mathematical models. We have developed a model for forecasting the number of Covid-19 cases and hospitalisations in Aotearoa New Zealand. The model works in two stages: firstly predicting the number of new cases and secondly estimating the proportion of those cases that will need hospital treatment. The model produces a range of likely values, which is important because is impossible to predict with 100% accuracy. We show that the model does a reasonably good job of predicting hospitalisations up to 3 weeks ahead. The model has been used by public health agencies in Aotearoa New Zealand to help with healthcare capacity planning.

**Data Availability Statement:** The source code and data used to produce the results and analyses presented in this manuscript are available from the Github repository: https://github.com/michaelplanknz/covid19_forecasting_public Data

on Covid-19 hospital occupancy are publicly available at https://github.com/minhealthnz/nz-covid-data/tree/main/cases.

**Funding:** This research was funded by a grant from the New Zealand Department of the Prime Minister and Cabinet and Ministry of Health to MJP, LW and OJM. The funders played no role in the methodology design, data analysis, preparation of the manuscript or decision to publish. The Ministry of Health was responsible for collecting and supplying the data analysed in the study. URL of funders' websites: https://www.dpmc.govt.nz/ https://www.health.govt.nz/.

**Competing interests:** The authors have declared that no competing interests exist.

## Introduction

New Zealand used a combination of border controls and public health and social measures to suppress or eliminate transmission of SARS-CoV-2 in 2020 and 2021. By the end of 2021, high vaccine coverage had been achieved, with around 90% of those aged over 12 years having had at least two doses of the Pfizer/BioNTech BNT162b2 vaccine. Up to this time, there had been only around 3 confirmed cases of Covid-19 per 1,000 people and 0.01 Covid-19 deaths per 1,000 people. In February 2022, the B.1.1.529 (Omicron) variant began to spread in the community and subsequently caused a series of large waves dominated by a series of subvariants [1, 2].

A range of epidemiological models have been used to provide situational awareness and policy advice to inform the New Zealand Government's pandemic response. These have primarily consisted of increasingly complex mechanistic models of transmission dynamics, including factors such as age structure, vaccination status [3], social contact networks [4], waning immunity [5], reinfection [6], dynamic behavioural change, new variants [2]. This level of detail requires making relatively strong assumptions on the mechanisms underlying observed dynamics and is hence most appropriate for scenario analysis, which does not aim to make accurate long-term predictions but rather to deliver insights into key mechanisms affecting epidemic dynamics and a systematic approach to exploring the likely consequences of alternative strategies or policy decisions.

Near-term forecasting is another use of epidemiological modelling, distinct from medium-term or long-term scenario analysis. Here the focus is on accurately predicting epidemic dynamics and consequent demand for acute healthcare services over a time horizon of a few weeks [7]. Assuming no dramatic changes in the mechanisms driving observed epidemic dynamics over the short term, higher-level models can be used, which summarise the combined effects of underlying transmission mechanisms in terms of coarse-grained parameters that can be empirically estimated. This class of model includes so-called 'semi-mechanistic' models, typically based on the renewal equation [8–11]. These models require fewer detailed assumptions and are less sensitive to parameter uncertainty and model mis-specification. On the other hand, they aim to maintain sufficient mechanism and flexibility to respond realistically to changing trends in epidemiological data and be fitted, evaluated, interpreted and updated in real-time. They also account for known lags affecting epidemiological data streams, such as delays from infection to symptom onset, testing, hospital admission or death [12, 13].

Some approaches to epidemic forecasting incorporate more mechanistic assumptions about transmission based on the standard susceptible-exposed-infectious-recovered (SEIR) epidemiological modelling framework [14, 15]. This allows the effect of immunity in reducing transmission rates to be explicitly accounted for, which may improve forecast performance. However, the downside of this is that it typically requires additional data or assumptions about, for example, case ascertainment rates, effectiveness of vaccine-derived and infection-derived immunity, and waning immunity [16, 17]. An advantage of a simpler approach is that the combined effect of immunity and other factors affecting the time-varying reproduction number, such as contact patterns and population heterogeneity, is inferred empirically from the data in real-time.

Some forecasting frameworks incorporate independent data, for example from behavioural surveys, about changes in the average number of contacts per person [17–19] or contact rates between different age groups [11, 20]. Such data can allow the effects of potential behavioural change and age structure to be built into forecasts and their associated uncertainty. However, this type of data is not available in Aotearoa New Zealand.

Aotearoa New Zealand currently lacks dedicated forecasting tools for Covid-19 and other infectious diseases [21]. In this study, we present a method for forecasting Covid-19 cases,

hospital admissions and hospital occupancy in Aotearoa New Zealand. The model has been developed specifically for New Zealand's Covid-19 surveillance systems and data collection and reporting standards. The method is being used operationally by Te Whatu Ora (Health New Zealand) with support from Precision Driven Health to provide intelligence to health planners around the country. The model is a semi-mechanistic model for disease transmission based on the renewal equation [8, 10]. We use a Bayesian particle filter approach [12] to estimate the time-varying reproduction number and forecast the number of reported cases. This is coupled with Gaussian process regression models for the distribution of cases across age groups and the age-specific case hospitalisation ratio to forecast hospital admissions. Hospital occupancy is estimated using empirical data on age-specific length of hospital stay. We evaluate model performance by comparing forecasts generated from data supplied on a specific date to subsequently reported data.

## Methods

### Data

We used Ministry of Health data on reported cases of Covid-19 in New Zealand between 25 January 2022 and 24 July 2023. The dataset contained unit record data on age, report date and, for a subset of cases, self-reported symptom onset date. For hospitalised cases, data was available on the admission date and the number of days for which the patient was receiving hospital treatment for Covid-19, referred to as length-of-stay. This dataset was generated by the Ministry of Health by linking self-reported positive test results (mostly from self-administered rapid antigen tests) with hospital data from the National Minimum Dataset (NMDS) and Inpatient Admissions (IP) database based on national health index (NHI) number. In this dataset, hospital admissions are categorised by the Ministry of Health as either Covid-19-related or incidental (i.e. those who had tested positive but were not being treated for Covid-19), using clinical codes (for NMDS) or health specialty (for IP). There are significant time lags in reporting this information and as a result the number of Covid-19-related admissions recorded on a given day can vary from one update to another, and is typically incomplete for the most recent 1–2 weeks of data (see Fig A in S1 Text).

We also accessed Ministry of Health data on the total number of confirmed Covid-19 patients occupying an admitted bed (hospital occupancy). These data are from the Daily Hospital Capacity survey, which provides a count of the total number of Covid-19 patients in hospital each day and as such does not suffer from any significant reporting lag or revisions to historical data. These data are publicly available at https://github.com/minhealthnz/nz-covid-data/tree/main/cases and updated daily.

From the unit record data, we calculated the number of daily reported cases and number of new daily hospital admissions in 10-year age bands (Fig A in S1 Text). Hospitalisations that were classified as "not Covid-19-related" were excluded. We estimated the number of daily discharges from hospital by assigning each hospitalised case a pseudo-discharge date using their admission date and Covid-19-related length-of-stay. This is an approximation because some patients may have been admitted for non-Covid-19-related treatment and were only treated for Covid-19 later during they stay. However, we only use discharge data for visual comparison of model outputs, not for model fitting or validation.

We calculated the onset-to-report distribution for cases that were reported in the 70 days prior to the date the data was supplied and had an onset date recorded (Fig B in S1 Text). Cases with report date more than 7 days prior or more than 14 days after onset date ($< 0.2\%$ of the cases that had onset date recorded) were excluded.

We calculated the Covid-19-related length-of-stay distribution in each 10-year age band and the report-to-admission distribution for cases reported between 56 and 126 days prior to the date the data was supplied (Fig B in S1 Text). We did not include cases reported less than 56 days prior to this date in these calculations because of lags in recording hospitalisation data and right-censoring of patients who had not yet been discharged. Cases with Covid-19-related length-of-stay longer than 56 days (approximately 0.5% of admissions) were excluded. Cases with admission date more than 7 days prior to report date were excluded. This was a significant proportion (approximately 7%) of admissions but it is likely that many of these were initially admitted for non-Covid-related treatment and were only later treated for Covid.

## Model

We used a model consisting of two components. The first was a semi-mechanistic disease transmission model that was fitted to data on reported daily cases. We used this to produce simulated time series for cases, which can be projected forwards in time. The second component was a hospitalisation model that we used to estimate the time-varying, age-specific case hospitalisation ratio (CHR). We then applied this to the simulated time series for cases to produce simulated time series for admissions and hospital occupancy.

Reported cases represent only a fraction of all infections due to the fact that the majority of cases are self-reported and intensive case finding and contact tracing programmes had been wound down by the time of the study period in 2022–23. It is likely that high rates of mild and asymptomatic infection and high levels of population immunity during the study period further reduced case ascertainment. We did not attempt to estimate the total number of infections, which is difficult to do without serological data or regular testing of a representative cohort [22, 23]. Instead, we estimated the case hospitalisation ratio directly based on the proportion of cases in each age group that were hospitalised. This variable will be influenced both by disease severity and by case ascertainment rates. However, the key output of interest (near-term forecast hospitalisations) is insensitive to these factors once the number of cases and the case hospitalisation ratio are known.

**Transmission submodel.**　We modelled the number of cases $I_t$ infected on day $t$ using a semi-mechanistic framework based on the renewal equation [8]

$$I_t \sim \text{Poisson}\left(R_t \sum_{s=1}^{n} I_{t-s} u_s\right),\tag{1}$$

where $R_t$ is the time-varying instantaneous reproduction number and $u_t$ is the probability mass function for the generation time distribution, assumed to be a discretised Weibull distribution with mean 3.3 days and standard deviation 1.3 days [24, 25]. The reproduction number was modelled as a Gaussian random walk

$$R_t \sim N\left(R_{t-1}, \sigma_R\right).\tag{2}$$

Reporting lags were accounted for via a distribution $v_t$ of times from infection date to report date. This was the convolution of the incubation period distribution, assumed to be a discretised Weibull distribution with mean 3.2 days and standard deviation 2.2 days [26–28], and the empirical onset-to-report distribution (see Data section above). The expected number of

cases reported on day $t$, in the absence of any day-of-the-week effect, was therefore

$$Z_t = \sum_{s=1}^{n} I_{t-s} \nu_s. \tag{3}$$

We modelled the number of observed cases on day $t$ as

$$C_t \sim \text{NegBin} \left( \mu = \omega_{i[t]} Z_t, k_c \right), \tag{4}$$

where $\omega_{i[t]}$ is an empirical day-of-the-week effect ($i = t \bmod 7$) and $k_c$ is a dispersion factor. The day-of-the week effect was estimated directly from the data as the relative difference between daily reported cases $\hat{C}_t$ and the seven-day rolling average over a 15 week period:

$$\omega_i = \frac{1}{N} \sum_{t \bmod 7 = i} \frac{\hat{C}_t}{\sum_{s=t-3}^{t+3} \hat{C}_s}, \tag{5}$$

where $N$ is the number of weeks terms in the sum.

We fitted the model to the time series of reported daily cases $\hat{C}_t$ using a bootstrap filter as follows [29]. We simulated $M$ realisations (or particles) of the stochastic model defined by Eqs (1)–(3) (i.e. $M$ particles), with each particle consisting of time series for $I_t$, $R_t$ and $Z_t$. At each time step, particle $j$ was assigned a weight $j$ using the likelihood of the observed value of $\hat{C}_t$ under the distribution in Eq (4). The population of $M$ particles was then resampled by drawing, with replacement, from the full set of particles with weights $j$. For time steps after the last available data point (i.e. the prediction period), each particle was simply simulated forwards in time according to Eqs (1)–(3) with no filtering.

We initialised the model over a period of $t_{\text{init}} = 20$ days by drawing $I_t$ from a Poisson distribution with mean equal to the number of observed cases $\hat{C}_{t+m}$, where $m$ is the mean infection to report time. The value of $R_t$ at the end of the initialisation period ($t = t_{\text{init}}$) was drawn from the estimated posterior for $R_t$ based on the values of $I_s$ for $s < t$ using the method of [8]. Model results were not sensitive to the initialisation period because all model simulations were initialised a minimum of 88 days prior to the forecast date.

**Hospitalisation submodel.**    To estimate hospitalisations, we fitted models for the distribution of new cases by age and for the CHR in each 10-year age band. We fitted the log-transformed ratio $r_{it} = \hat{C}_{it}/\hat{C}_{i't}$ of cases in age group $i$ to cases in a reference age group $i'$ (arbitrarily set to be the 40–49-year group), and the logit-transformed CHR in age group $i$ as independent Gaussian processes over time:

$$\log(r_{it}) \quad \sim \quad GP(\mu(t), K(t, t')), \tag{6}$$

$$\text{logit } \text{CHR}_{it} \quad \sim \quad GP(\mu(t), K(t, t')), \tag{7}$$

where $\text{CHR}_{it}$ was defined as the proportion of cases reported on day $t$ that were hospitalised for Covid-19 using a 7-day centred rolling average. These models were trained using the *fitrgp* package in Matlab2022b with a squared exponential kernel $K$ and default hyperparameter settings. We fitted the age distribution model to data in the 56 days prior to the most recent available data. To allow for reporting lags in hospitalisation data, we fitted the CHR model to data between 84 and 21 days prior to the most recent available data.

We then used the fitted models to predict the overall CHR on day $t$ as

$$\text{CHR}_t = \frac{\sum_i r_{it} \text{CHR}_{it}}{\sum_i r_{it}} \qquad (8)$$

We included model uncertainty in $r_{it}$ and $\text{CHR}_{it}$ by independently sampling different trajectories from the fitted Gaussian processes for each particle $j$.

We then simulated the number of new admissions $A_t$ on day $t$ by applying the predicted CHR from Eq (8) to the output $I_t$ of the particle filter:

$$A_t \sim \text{NegBin}\left(\mu = A_t^* \text{CHR}_t, k_h\right), \qquad (9)$$

where $k_h$ is a dispersion factor, $A_t^* = \sum_t I_{t-s} w_s$ and $w_s$ is the probability mass function for the distribution of time from infection to admission. This distribution was estimated as the convolution of the assumed distribution for the time from infection to report $\nu_t$ and the empirical distribution for the time from report to admission.

In order to predict hospital occupancy, we also needed to model hospital discharges. We modelled the distribution of Covid-19-related length-of-stay for cases admitted on day $t$ by combining the empirical age-specific length-of-stay distributions with the modelled age distribution of hospitalised cases. Specifically, the probability $l_{st}$ that an admission on day $t$ will have length-of-stay $s$ days was calculated as

$$l_{st} = \frac{\sum_i l_{si}^{(\text{age})} r_{it} \text{CHR}_{it}}{\sum_i r_{it} \text{CHR}_{it}}, \qquad (10)$$

where $l_{si}^{(\text{age})}$ is the probability that an admission in age group $i$ will have length-of-stay $s$ days.

We calculated the number of discharges $D_t$ on day $t$ by summing over day of admission $t'$:

$$D_t = \sum_{t'} N_{t-t',t'}, \qquad (11)$$

where $N_{st} \sim \text{Multinomial}(A_t, l_{st})$ is the number of admissions on day $t$ that have Covid-19-related length-of-stay $s$. We calculated net change in hospital occupancy since day $t_0$ as the cumulative number of admissions minus the cumulative number of discharges since day $t_0$. Hospital occupancy at time $t = t_0$ was set so that the mean and standard deviation of the particles' hospital occupancy were equal to the observed hospital occupancy on day $t_0$ and the standard deviation in observed hospital occupancy in the week prior to $t_0$ respectively.

**Forecast generation and evaluation.**   In order to test the performance of the model against out-of-sample data, we generated forecasts using data supplied on one of a series of dates spaced at one-week intervals from 2 October 2022 to 23 July 2023. This ensured that forecasts were based only on the data that was available at a given time point, at which recent hospitalisation data was typically incomplete (Fig A in S1 Text). We then compared forecasts generated at time $t_f$ with subsequently observed data at times $[t_f - 6, t_f]$ (nowcast), $[t_f + 1, t_f + 7]$ (7-day forecast), $[t_f + 8, t_f + 14]$ (14-day forecast), and $[t_f + 15, t_f + 21]$ (21-day forecast).

We quantified forecast skill by calculating the continuous ranked probability score (CRPS) (see e.g. [30]) and bias. For a forecast specified by cumulative distribution function $F(x)$ and data $\hat{x}$, the CRPS is defined as

$$\text{CRPS}(\hat{x}) = \int_{-\infty}^{\infty} \left(F(x) - I(x \geq \hat{x})\right)^2 dx \qquad (12)$$

where $I(.)$ is the indicator function. Bias is defined as $\text{bias}(\hat{x}) = 1 - 2F(\hat{x})$. This metric lies

**Table 1. Parameter values used in the model.**

| Parameter | Value |
|---|---|
| Generation time mean (s.d.) | 3.3 (1.3) days |
| Incubation mean (s.d.) | 3.2 (2.2) days |
| Std. dev. in daily random walk step for $R_t$ | $\sigma_R = 0.025$ |
| Dispersion factor for daily cases | $k_c = 100$ |
| Dispersion factor for daily admissions | $k_h = 100$ |
| Number of particles | $M = 10^5$ |
| Initialisation period for renewal equation model | $t_{\text{init}} = 20$ days |

between −1 and 1 and is equal to zero of the data coincides with the median of the forecast distribution.

We calculated the CRPS on log transformed data using the transformation $\tilde{x} = \ln(x + 1)$. This better reflects the exponential nature of epidemic growth and decay, and leads to CRPS values that are independent of the magnitude of the observed quantity [31], which will be very different for cases compared to admissions for example. It also means the CRPS values can be interpreted as a probabilistic measure of relative error [31]. For example, for a point forecast $x_f$, it follows from Eq (12) that

$$\exp(\text{CRPS}) - 1 = \frac{|x_f - \hat{x}|}{\min(x_f, \hat{x}) + 1} \tag{13}$$

which is an approximation to the relative difference between the forecast $x_f$ and the data $\hat{x}$.

Model parameters are shown in Table 1. The chosen values for the random walk standard deviation $\sigma_R$ and dispersion factors $k_c$ and $k_h$ were found to give a reasonable balance between being responsive to changes in trends while avoiding overfitting (see Results section for sensitivity analysis). Note that for the chosen values of $k_c$ and $k_h$, the distributions for daily cases and daily admissions are close to Poisson, which is the limiting case of a negative binomial distribution as $k \to \infty$. However, we retained the more general negative binomial model to provide flexibility in modelling other datasets, which may have higher variation in these quantities. Data and documented code to reproduce the results are available at https://github.com/michaelplanknz/covid19_forecasting_public.

## Results

Fig 1 shows the fitted Gaussian process regression models for the age distribution of reported cases and the age-specific case hospitalisation ratio (CHR). The models were fitted to data supplied on an example forecast date (16 April 2023) and then projected forwards in time and compared to subsequently available data up to 7 May 2023. Overall, the fitted models made good predictions for future, out-of-sample data, which generally fell within the 95% prediction intervals and visually exhibited a similar level of temporal autocorrelation as simulated model trajectories. There were some notable exceptions. For example the proportion of cases in age bands in the under 20 years and 60 to 80 years range started to track outside the predicted intervals 2–3 weeks after the forecast date (Fig 1a). The CHR in the 0–10 years age group deviated outside the prediction interval for a period of time around 1–2 weeks after the forecast date (Fig 1b). The fitted regression model for CHR in the 40–50 years age group represented the data as a lower frequency signal with higher noise relative to the other age groups.

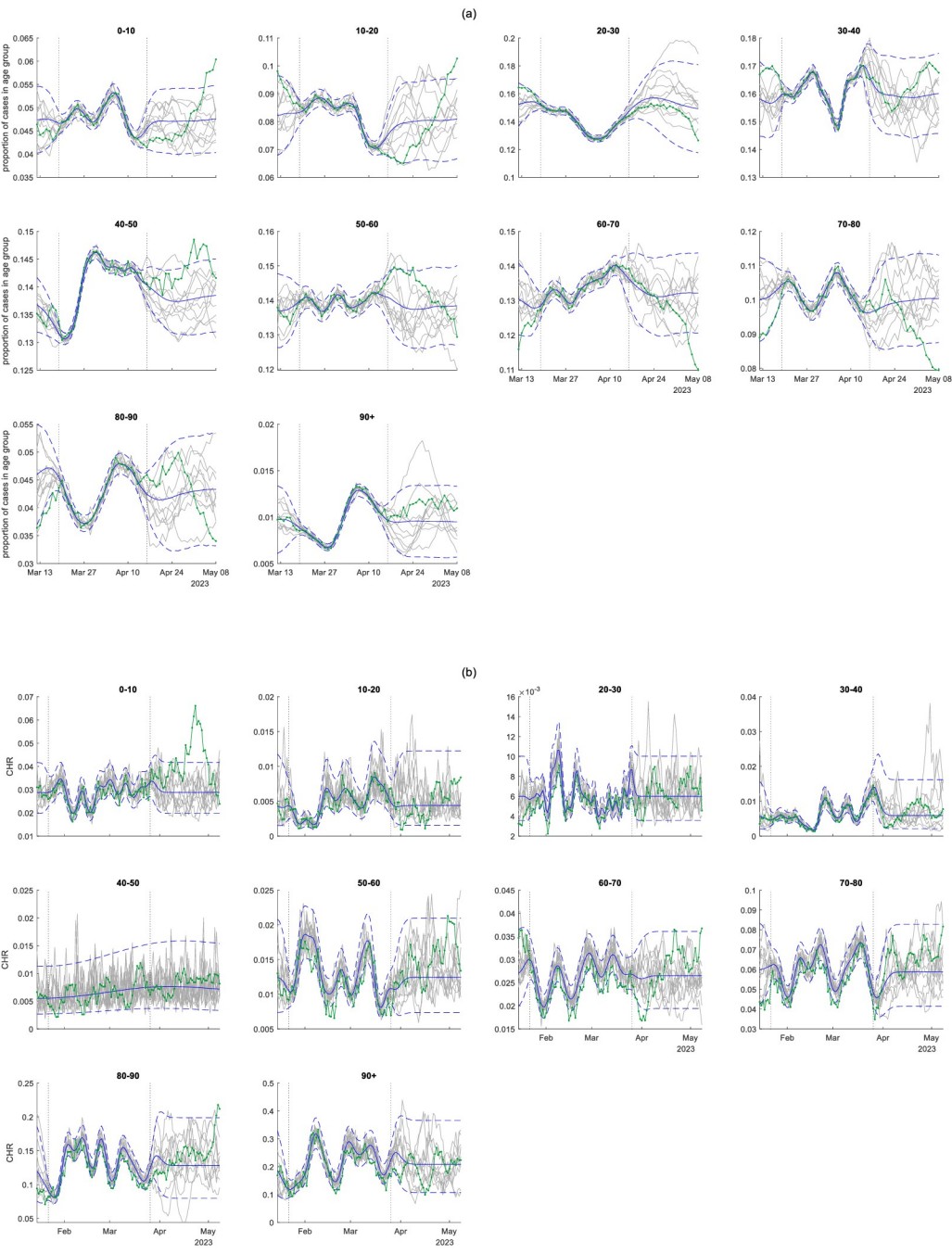

**Fig 1.** Fitted Gaussian process regression models for: (a) the proportion of cases in each age group; (b) the case hospitalisation ratio (CHR) in each age group. Models fitted to data supplied on 16 April 2023. Dotted vertical lines shown the fitting window (19 March to 16 April 2023 for proportion of cases in each age group, 22 January to 26 March 2023 for CHR). Each panel shows the mean (solid blue) and 95% prediction interval (dashed blue) of the fitted model, ten example simulated trajectories from the fitted model (grey), and comparison to subsequently observed data up to 7 May 2023 (green).

Fig 2 shows the forecast for cases, new hospital admissions and hospital occupancy, generated from data supplied on 16 April 2023. The forecast performed reasonably well when compared against subsequently available data up to 7 May 2023. Daily cases and daily admissions were almost entirely within the 90% prediction interval, although predominantly below the predicted median, indicating that epidemic growth slowed in the 3 weeks following the forecast date. Note that the historic modelled levels of hospital occupancy (to the left of the dotted vertical line in Fig 2d) may deviate from the data because the model occupancy was not produced by fitting directly to hospital occupancy data, but by calculating net change in hospital occupancy relative to the forecast date from simulated daily admissions and discharges. Therefore, unlike cases and admissions, accuracy in modelled occupancy tends to decrease the further backwards in time you go relative to the forecast date. However, this is not important for the purposes of forecasting.

In order to assess forecast performance over time, Fig 3 shows the full time series of data alongside the results of the forecast that was generated between 15 and 21 days previously (data available on the third Sunday prior). The accuracy of the forecast 15–21 days ahead was variable but the large majority of data points (89% for cases, 87% for admissions and 85% for occupancy) fell within the 90% prediction interval. The most notable deviation is that the forecast overestimated the height of the peak that occurred in December 2022. This may be partly explained by a drop-off in testing and reporting of cases during the Christmas summer holiday period, as indicated by wastewater surveillance [32], and other holiday-related effects on transmission rates. The forecast also overestimated hospitalisations during this period, but to a

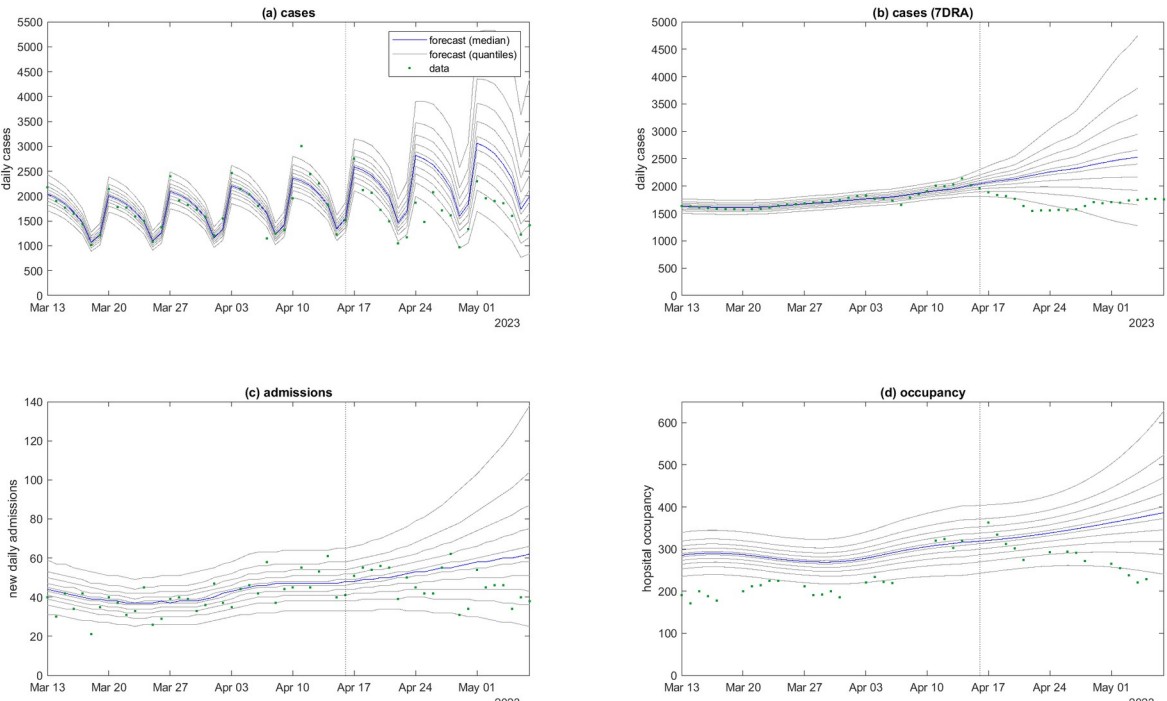

**Fig 2.** Model fitted to data up to 16 April 2023 (vertical dotted line), projected forwards in time for 21 days and compared to subsequently available data available up to 7 May 2023 for: (a) new daily cases; (b) smoothed daily cases (seven-day rolling average); (c) new daily hospital admissions; (d) hospital occupancy. The day-of-the-week effect is visible in panel (a) for reported daily cases. Blue curve is the median and grey curves are the 5th, 15th, . . ., 85th, 95th percentiles of $M = 10^5$ particles, calculated by drawing one sample per particle from the distributions specified by Eqs (4), (9) and (11).

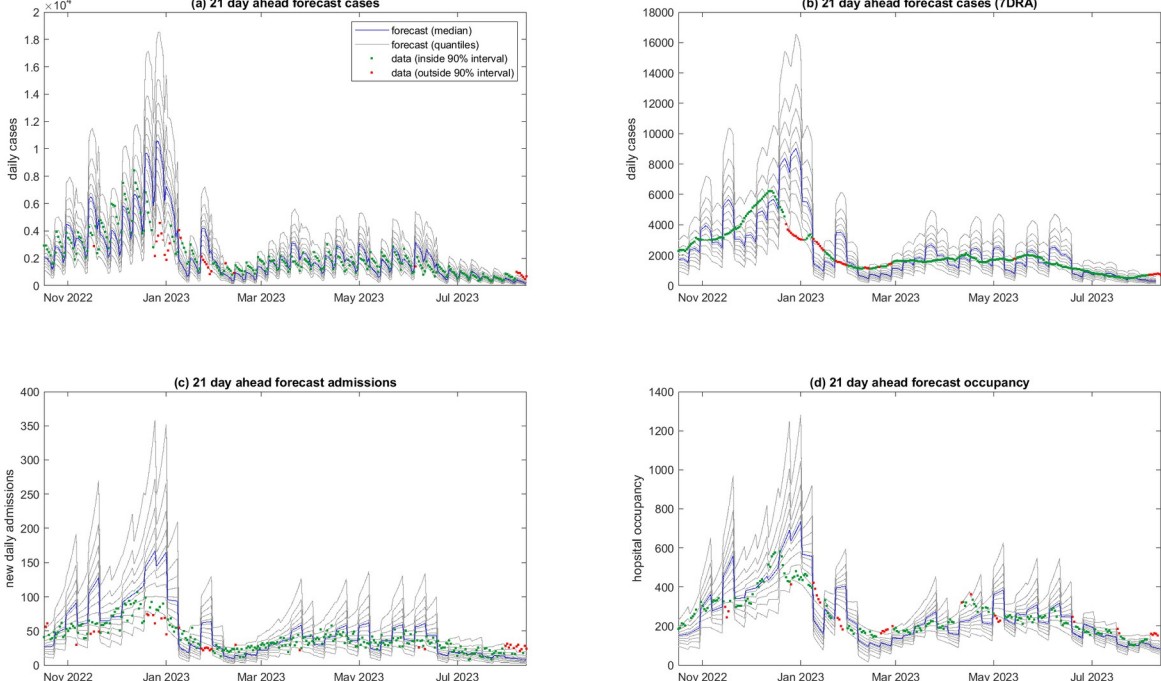

**Fig 3. 21-day ahead forecast performance.** Model results generated from data supplied at one of a series of weekly time points from 2 October 2022 to 23 July 2023, compared to actual data for the period 15–21 days subsequent to the date the data was supplied for: (a) new daily cases; (b) smoothed daily cases (seven-day rolling average); (c) new daily hospital admissions; (d) hospital occupancy. Testing data was supplied on 20 August 2023 (i.e. 4 weeks subsequent to the last forecast). Weekly discontinuities in the forecasts are because each 7-day block represents a forecast generated from data supplied on a different date. Blue curve is the median and grey curves are the 5th, 15th, . . ., 85th, 95th percentiles of $M = 10^5$ particles. Data points outside the 5th –95th percentile range of the forecast are shown in red.

lesser extent than it overestimated cases. Accurately predicting the peak of a wave is known to be a difficult problem in epidemic forecasting [33], and other models have suffered from similar problems [15, 17].

Forecast skill was generally higher for a shorter time horizon. For example, in the 7-day ahead forecast (see Fig C–D in S1 Text), the prediction intervals were more tightly focused around the subsequent data in most cases compared to the 21-day ahead forecast. In general, the CRPS increased with time (Fig 4a), indicating that forecast accuracy deteriorated as the time horizon was extended. For a 3-week time horizon, the mean CRPS on log transformed data was approximately 0.25 for cases and admissions and around 0.17 for occupancy.

The admissions forecast was positively biased, particularly at short time horizons, whereas the occupancy forecast was negatively biased although less strongly (Fig 4b). The case forecast was close to unbiased. This suggests that the model may be overestimating the case hospitalisation ratio. This could be a consequence of the imperfect fit of the Gaussian process regression, for example due to a non-normal distribution of the CHR over time.

Model results were not highly sensitive to the chosen values of the random walk standard deviation $\sigma_R$. Smaller values of $\sigma_R$ meant the model was less able to capture rapid changes in the reproduction number. This tended to lead to less accurate forecasts, with fewer data points falling inside the 90% prediction intervals (71%, 78% and 69% of data points for cases, admissions and occupancy respectively)—see Fig E in S1 Text. Higher values of $\sigma_R$ introduced more uncertainty into forecasts, meaning that prediction intervals were unnecessarily wide (Fig F in S1 Text). For the chosen values of the dispersion factors $k_c$ and $k_h$ (see Table 1), the

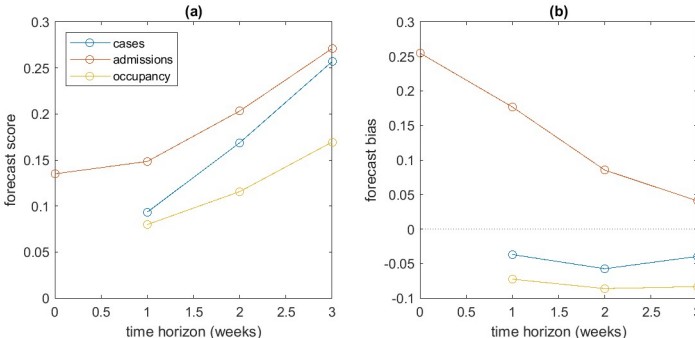

**Fig 4.** Forecast performance quantified by (a) continuous ranked probability scores (CRPS), and (b) bias, for forecasts up to 0, 1, 2 and 3 weeks ahead. Model results generated from data supplied at one of a series of weekly time points from 2 October 2022 to 23 July 2023, and tested against data supplied on 20 August 2023. Smaller scores indicate more accurate forecasts, and values of bias closer to zero indicated less biased forecasts.

distributions of daily reported cases and daily hospital admissions were close to Poisson and this produced a reasonable fit for the current dataset. Smaller values of $k_c$ and $k_h$ resulted in prediction intervals that were too wide for the modelled dataset (Fig G in S1 Text).

We did not encounter issues with particle degeneracy in the bootstrap filter (see Fig H in S1 Text), which is a problem that can arise when almost all particles have nearly zero weight and so the number of unique particles becomes very low [29]. It is possible this could be an issue if the model were run for a longer period of time, or if there were more noise or abrupt changes in the data. This could require tuning or fitting of the dispersion parameter $k_c$.

## Discussion

Near-term forecasting of infectious disease activity and consequent demand for acute health-care can support situational awareness, planning and public health response [7]. We have developed a method for forecasting Covid-19 cases, hospital admissions and hospital occupancy based on Aotearoa New Zealand's unique disease surveillance and data collection systems. The method couples a semi-mechanistic model for disease transmission to forecast cases with Gaussian process regression models for the time-varying case hospitalisation ratio.

We have demonstrated that the model provides useful forecasts by benchmarking against subsequently observed data up to 21 days ahead. The forecasting tool has been operationalised by Te Whatu Ora Health New Zealand in 2023 to provide weekly national and regional level forecasts in real-time. Our method is a useful component of health system capacity planning and response to Covid-19. It is also an important step towards development of more sophisticated situational awareness and forecasting capability in Aotearoa New Zealand for other infectious diseases and future pandemic threats.

Strengths of our model include that it is specifically designed to use New Zealand's unique Covid-19 data streams, including linked unit record data on date of symptom onset and case report and, where applicable, date of hospital admission and length of stay. This data allowed us to empirically estimate the distribution of onset-to-report time, report-to-admission time and age-specific length of hospital stay. The model accounts for known lags in the reporting of hospital admissions for Covid-19, but uses up-to-date data on reported cases in each age group to improve accuracy of hospitalisation forecasts. Forecasts performed reasonably well when benchmarked against subsequently observed, out-of-sample data.

Numerous variables affect the age-specific case hospitalisation ratio (CHR), such as vaccine coverage, rates of prior infection, comorbidities and case ascertainment (which affects the denominator of the ratio) [16]. Our method avoids the need for assumptions about the effects of these variables by taking an empirical approach to estimating the age-specific case hospitalisation ratio from recent data. This is reasonable because, although the variables affecting CHR will vary over time, they will generally vary slowly relative to the typical forecasting time horizon of 1–3 weeks.

The model has several important limitations. It assumes that, over the forecasting time horizon, the effective reproduction number follows a simple random walk. This ignores mechanisms that may systematically affect transmission dynamics (e.g. depletion of the susceptible population, changes in contact patterns) meaning it is not suitable for forecasting more than a few weeks ahead, and cannot provide any insight into the reasons for changes in transmission patterns or the effects of possible interventions. Our results show that although the forecast is reasonably accurate for the week ahead, accuracy deteriorates for two-week and three-week forecasts. This to be expected, but is important to remember in practical applications of the model and suggests that forecasts should be regularly updated with the most recent available data.

Abrupt changes in case ascertainment, for example as the result of a policy change or a change in access to testing, would invalidate the forecast for a period of time until the time window used for estimating the CHR falls inside the new case ascertainment regime. The same would apply if there was an abrupt change in clinical severity, for example due to rapid take-over of a new variant. Other than a temporary drop in case ascertainment during the 2022–2023 holiday period [32], there is no evidence of these issues arising during the study period of October 2022 to July 2023. However, there was subsequently an abrupt drop in case ascertainment following the lifting of the government isolation mandate for confirmed cases of Covid-19 on 14 August 2023.

We have applied and tested the model in Aotearoa New Zealand during a period in which the Omicron variant of SARS-CoV-2 was dominant, there were limited non-pharmaceutical interventions, and increasing levels of hybrid immunity [1, 2, 34]. Application of the model in other contexts, such as in an immune naive population or during periods of intense non-pharmaceutical interventions or behavioural change, would likely require significant model adaptation and recalibration.

An alternative approach to forecasting cases and modelling hospitalisations as a time-varying fraction of cases would be to model the trend in hospitalisations directly. This would have the advantage of avoiding issues arising from changes in case ascertainment rates, and may be necessary in future as case ascertainment declines. However, our approach has the advantage that cases are a leading indicator compared to hospital data, which is significantly lagged due to both the lag in onset of severe illness and the hospital data reporting lag. Therefore, changes in trends in transmission dynamics or in the age distribution of infections immediately feed into the forecast for hospitalisations.

Future improvements of the model could incorporate wastewater surveillance data as an independent measure of prevalence [32] and more mechanistic transmission assumptions, for example to account for the accumulation of population immunity during a wave or following a vaccine rollout.

## Supporting information

**S1 Text. Supporting text and supplementary figures.**
(PDF)

## Acknowledgments

The authors acknowledge the role of the New Zealand Ministry of Health, Te Whatu Ora Health New Zealand, and the Institute of Environmental Science and Research in supplying data in support of this work. The authors are grateful to Nicholas Kondal and Rachel Owens at Precision Driven Health for code validation and automating data pipelines, to James Harris at Te Whatu Ora for discussions about model methods and interpretation, to Rob Moss for useful discussions on forecasting models, and to Fiona Callaghan and the Covid-19 Modelling Government Steering Group for feedback on earlier versions of this work.

## Author Contributions

**Conceptualization:** Michael J. Plank.

**Data curation:** Michael J. Plank.

**Formal analysis:** Michael J. Plank.

**Funding acquisition:** Michael J. Plank.

**Investigation:** Michael J. Plank, Leighton Watson, Oliver J. Maclaren.

**Methodology:** Michael J. Plank, Leighton Watson, Oliver J. Maclaren.

**Project administration:** Michael J. Plank.

**Software:** Michael J. Plank.

**Validation:** Michael J. Plank.

**Visualization:** Michael J. Plank.

**Writing – original draft:** Michael J. Plank.

**Writing – review & editing:** Michael J. Plank, Leighton Watson, Oliver J. Maclaren.

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
