## [Decision Letter · Decision Letter 0]

13 Nov 2023

Dear Dr Plank,

Thank you very much for submitting your manuscript "Near-term forecasting of Covid-19 cases and hospitalisations in Aotearoa New Zealand" for consideration at PLOS Computational Biology. As with all papers reviewed by the journal, your manuscript was reviewed by members of the editorial board and by several independent reviewers. The reviewers appreciated the attention to an important topic. Based on the reviews, we are likely to accept this manuscript for publication, providing that you modify the manuscript according to the review recommendations.

Sincerely,

Alex Perkins

Academic Editor

PLOS Computational Biology

Virginia Pitzer

Section Editor

PLOS Computational Biology

Reviewer's Responses to Questions

**Comments to the Authors:**

Reviewer #1: The paper describes and evaluates a methodology for forecasting COVID-19 cases and hospital occupancy in Aotearoa New Zealand. The description of the model is comprehensive and concise. The treatment of input data, including the exclusion of censored data points is well described. The use of CRPS and bias scores to measure forecasting performance is appropriate and subsequent results well presented. I note a few points of suggestion that may improve the work as it stands:

1. The magnitude and pattern of data censoring in Supplementary Figure 1b is hard to discern due to the small size of the plot. Increasing the size of this plot (or otherwise changing the data presentation) would help clarify this.

2. Line 146 – “of cased” to “of cases”

3. Line 190 – “as independent Gaussian process” to “as independent Gaussian processes” (or “as an independent Gaussian process”?)

4. Discussion of the calibration of forecast outputs (e.g. line 263, 275) would be strengthened by the inclusion of a result quantitatively evaluating this (e.g. reporting the percentage of observations within the 90% intervals or a probability integral transform plot [1]).

[1] Diebold, F. X., Gunther, T. A., & Tay, A. S. (1998). Evaluating density forecasts with applications to financial risk management. International Economic Review, 39(4), 863.

Reviewer #2: This paper was easy to read and straightforward. It’s nice to see this type of model in use in a public health setting. The comments I’m providing are mostly suggestions and not “make or break.” Thanks!

Major comments

- Did you all consider modeling the hospital admissions and occupancy directly, instead of modeling them as a portion of cases? The only reason I bring this up is because that type of data is much more reliable than case data and I’d imagine it would be more straightforward to model them directly.

- Did you have any issues with particle convergence (i.e., where the effective number of particles declines to some nominal number due to large differences in likelihood-based weights)? How did you deal with those? If not, how did you avoid particle convergence?

- I was wondering a bit on the rationale for some of the parameter choices in Table 2. Specifically with the dispersion parameter values being quite high. Did you consider just using a Poisson model instead of a Negative Binomial model for cases w/ a dispersion parameter of 100? If you have any testing that you performed in picking the parameter values, it would be interesting (although not necessary) to see examples of how the model results differ with lower dispersion parameters.

Minor comments

- For figure 3, could you please add in the legend what the red color denotes?

- For figure 2 and 3 (and any w/ that same set up), I think it would be more useful to see the uncertainty visualized as a shading gradient rather than lines for the quantiles. Additionally, I don’t think it’s necessary to have so many quantiles shown, perhaps showing just 5, 25, median (as a thicker line), 75, 95 would be good enough? The way that those figures are set up right now are busy and somewhat difficult to analyze.

- In the discussion, it would be useful to comment more on instances where you think the model failed or did poorly on the forecast scoring. For example, why do you think the hospital admissions and hospital occupancy models are positively and negatively biased? What does that mean practically speaking and how does it affect the interpretation of the results?

Reviewer #3: The paper presented a forecasting model suited to New Zealand's unique COVID-19 surveillance systems. This combined a semi-mechanistic stochastic model of cases and a Gaussian process regression model of the age-specific case hospitalisation ratio. Hidden states related to cases were fit using a particle filter, age-specific lengths of stay in hospital were directly estimated empirically, relative number of cases by age group and case hospitalisation ratio through time were fit to Guassian process regression models.

The approach to modelling is sound for the forecast horizon of 3 weeks (noting that susceptible depletion will play a role over longer time horizons) and the model provides reasonable forecasts of New Zealand daily cases and hospital admissions. The approach here is nice in that it present a flexible model that avoids the added complexity of features such as vaccination or past waves of infection. I have some minor suggestions for improvements, but otherwise this paper is appropriate for publication.

(1) Figure 2 is confusing, the authors describe the particles as relating to states I, R and Z and these states being fit to case data, so quantiles for the case trajectory do not make sense in regions where there is data. My guess is the authors have sampled from the relevant negative binomial distribution given Z_t's of each particle, but this is not described.

(2) In Figure 1(b) the estimated 40-50 age group case hospitalisation ratio does not track the data particularly well in comparison to the other age groups. Some comments on this in the text would be helpful.

(3) Pg 6. has typographical error 'cased'

(4) Pg 7. Z_t is described as the 'expected number of cases reported on day t' but this isn't quite accurate as Z_tw_{i[t]} is the expected number of cases on day t

(5) Pg8. N is not defined

(6) Pg9. Equation 11 should not have the A_{t'}

**Have the authors made all data and (if applicable) computational code underlying the findings in their manuscript fully available?**

Reviewer #1: Yes

Reviewer #2: Yes

Reviewer #3: None

PLOS authors have the option to publish the peer review history of their article (what does this mean?). If published, this will include your full peer review and any attached files.

Reviewer #1: **Yes: **Ruarai Tobin

Reviewer #2: No

Reviewer #3: No

Figure Files:

Data Requirements:

Reproducibility:

References:

---

## [Editor Report · Decision Letter 1]

12 Dec 2023

Dear Dr Plank,

We are pleased to inform you that your manuscript 'Near-term forecasting of Covid-19 cases and hospitalisations in Aotearoa New Zealand' has been provisionally accepted for publication in PLOS Computational Biology.

Best regards,

Alex Perkins

Academic Editor

PLOS Computational Biology

Virginia Pitzer

Section Editor

PLOS Computational Biology

---

## [Editor Report · Acceptance letter]

27 Dec 2023

PCOMPBIOL-D-23-01527R1 

Near-term forecasting of Covid-19 cases and hospitalisations in Aotearoa New Zealand

Dear Dr Plank,

I am pleased to inform you that your manuscript has been formally accepted for publication in PLOS Computational Biology. Your manuscript is now with our production department and you will be notified of the publication date in due course.

With kind regards,

Zsofi Zombor
